# The Cellular Senescence Stress Response in Post-Mitotic Brain Cells: Cell Survival at the Expense of Tissue Degeneration

**DOI:** 10.3390/life11030229

**Published:** 2021-03-11

**Authors:** Eric Sah, Sudarshan Krishnamurthy, Mohamed Y. Ahmidouch, Gregory J. Gillispie, Carol Milligan, Miranda E. Orr

**Affiliations:** 1Department of Internal Medicine, Section on Gerontology and Geriatric Medicine, Wake Forest School of Medicine, Winston-Salem, NC 27157, USA; esah@wakehealth.edu (E.S.); sukrishn@wakehealth.edu (S.K.); ahmimy17@wfu.edu (M.Y.A.); ggillisp@wakehealth.edu (G.J.G.); 2Bowman Gray Center for Medical Education, Wake Forest School of Medicine, Winston-Salem, NC 27101, USA; 3Departments of Biology and Chemistry, Wake Forest University, Winston-Salem, NC 27109, USA; 4Sticht Center for Healthy Aging and Alzheimer’s Prevention, Wake Forest School of Medicine, Winston-Salem, NC 27157, USA; 5Department of Neurobiology and Anatomy, Wake Forest School of Medicine, Winston-Salem, NC 27157, USA; milligan@wakehealth.edu; 6Salisbury VA Medical Center, Salisbury, NC 28144, USA

**Keywords:** cellular senescence, post-mitotic, neuronal senescence, Alzheimer’s disease, biology of aging, neurodegeneration, brain, geroscience, amyotrophic lateral sclerosis, tauopathy

## Abstract

In 1960, Rita Levi-Montalcini and Barbara Booker made an observation that transformed neuroscience: as neurons mature, they become apoptosis resistant. The following year Leonard Hayflick and Paul Moorhead described a stable replicative arrest of cells in vitro, termed “senescence”. For nearly 60 years, the cell biology fields of neuroscience and senescence ran in parallel, each separately defining phenotypes and uncovering molecular mediators to explain the 1960s observations of their founding mothers and fathers, respectively. During this time neuroscientists have consistently observed the remarkable ability of neurons to survive. Despite residing in environments of chronic inflammation and degeneration, as occurs in numerous neurodegenerative diseases, often times the neurons with highest levels of pathology resist death. Similarly, cellular senescence (hereon referred to simply as “senescence”) now is recognized as a complex stress response that culminates with a change in cell fate. Instead of reacting to cellular/DNA damage by proliferation or apoptosis, senescent cells survive in a stable cell cycle arrest. Senescent cells simultaneously contribute to chronic tissue degeneration by secreting deleterious molecules that negatively impact surrounding cells. These fields have finally collided. Neuroscientists have begun applying concepts of senescence to the brain, including post-mitotic cells. This initially presented conceptual challenges to senescence cell biologists. Nonetheless, efforts to understand senescence in the context of brain aging and neurodegenerative disease and injury emerged and are advancing the field. The present review uses pre-defined criteria to evaluate evidence for post-mitotic brain cell senescence. A closer interaction between neuro and senescent cell biologists has potential to advance both disciplines and explain fundamental questions that have plagued their fields for decades.

## 1. Introduction

Many debilitating diseases affecting our modern population have resulted from the deterioration of biological processes suited for a 40-year lifespan. Exceptional examples are neurodegenerative diseases. Age is the single greatest risk factor for the most common neurodegenerative diseases including Alzheimer’s disease (AD), Parkinson’s disease (PD), and amyotrophic lateral sclerosis (ALS). Sporadic neurodegenerative diseases, i.e., those not inherited, rarely affect adults before the age of 50 and are nearly absent in adults younger than 40-years-old. In this way, evolution strongly favored nervous system health. Indeed, adaptation accounts for the success of humans in our recent history, a behavioral flexibility dependent on the nervous system. For these reasons neurodegenerative diseases and central nervous system (CNS) injuries (stroke) are devastating because of the limited regenerative capacity of the mature tissues.

The complexity of the structure and function of the nervous system is achieved through extensive developmental processes and is maintained in part due to the resiliency of the cells to maintain function and resist activating cell death processes. This first evidence of this phenomenon was reported by Rita Levi-Montalcini and Barbara Booker in 1960 in their pioneering experiments demonstrating the critical role of nerve growth factor (NGF) in sympathetic neuron growth and survival [1]. Applying NGF antiserum to newborn mouse neurons resulted in a 97–99% cell loss; the same strategy only resulted in a 34% neuron loss in the adult mouse [1]. Less differentiated cells that fail to interact with their target appear to die by a morphological process (nuclear cell death) that we can now attribute to apoptosis [2]. Apoptosis is an effective and efficient mechanism of cell death involving robust activation of caspases. More mature neurons appear to die in a slower process that was termed cytoplasmic with prominent changes occurring in mitochondria, endoplasmic reticulum (ER), and lysosomes. Several forms of neuronal cell death have been described [3]. Reactivation of cell cycle in post-mitotic neurons has also been reported to be an initiating event for neuronal death (reviewed in [4]). The alterations in mitochondrial function, fission and fusion, ER stress, protein misfolding and aggregations, autophagy, and expression of proteins associated with cell cycle regulators are observed in neurodegenerative diseases and assumed to be processes associated with the neuron’s cell death. However, in a post-mitotic system, cellular evolution may have favored a pro-survival response for difficult-to-replace cells in post-mitotic tissues that simultaneously prevented malignancy in mitotically competent cells–senescence.

In 1961, Leonard Hayflick and Paul Moorhead reported a dogma-shifting observation from their cell culture experiments. At that time, primary cells grown in culture were believed to be immortal with indefinite replicative potential. However, Hayflick and Moorhead reported cessation of growth and eventual loss of the lines routinely after ~50 passages or one year in culture. They referred to the phenomenon as “senescence at the cellular level” and hypothesized that it was attributed to intrinsic factors [5]. Subsequent studies revealed that human cells track their cell divisions. Cultured human fibroblasts replicate 50–80 times and then no longer divide, which is referred to as replicative senescence or the “Hayflick limit” [6,7]. By mathematical definition, a “limit” can be approached but not achieved, which is ironically fitting for this phenomenon-translating these in vitro observations to tissues and living organisms have been proposed by many, but a consensus definition has not been reached. Toward this end, we frame senescence as a complex stress response that culminates a change in cell fate.

Replicative senescence, as defined by Hayflick and Moorhead, has clearly defined underlying biology (telomere attrition) and functional phenotypes (inability to divide). Subsequent studies have identified exogenous factors that can cause cell cycle arrest through telomere-independent mechanisms. As the number of exogenous senescence-inducing factors has expanded, so has the number of unique cell types being interrogated. Numerous phenotypes have emerged as a result. Senescent cells have been identified using various morphology markers; gene, protein, metabolic changes; and functional readouts and have been a subject of earlier reviews [8,9,10,11]. A specific combination of phenotypes defining senescence currently does not exist [12]; however, most agree that it is a stress-induced change in cell fate which includes a stable cell cycle arrest and cell death resistance.

The identity of the parent cell type and upstream signals has consequences on the post-senescence phenotype [13,14]. The resulting heterogeneity has presented challenges for identifying, defining, and studying senescent cells in vivo and across disciplines. Where biologists agree is that interpreting the senescence phenotype requires integrating various lines of distinct evidence placed in appropriate context. This is especially true for post-mitotic tissues such as the brain. The intent of this literature review is not to list studies that used an umbrella term “senescence” to describe a physiological response. Rather, it is to critically evaluate results from reports on brain cell senescence using pre-defined senescence-defining criteria: proliferative/cell cycle arrest, apoptosis resistance, senescence-associated secretory phenotype (i.e., cytokines, chemokines, pathogenic proteins, exosomes, miRNA, enzymes, etc.), and senescence associated b-galactosidase activity (SA β-gal) (Figure 1). The International Cell Senescence Association (ICSA) recently provided a list of key cellular and molecular features of senescence [9]. They acknowledged that post-mitotic cells may develop features of senescence, however, their recommendations largely focused on dividing cells. Some of which have major limitation when evaluating brain tissue (i.e., lipofuscin and SA β-gal). Here we provide an overview of senescent phenotypes, as relevant to post-mitotic cells, the assays used to assess them, and the markers relevant to senescence, each in the context of brain cell biology. We then review different brain cell types using these criteria. Overall, the present contribution aims to provide an accessible summary on senescent post-mitotic brain cells with criteria and interpretations relevant to the neurobiology of aging and disease.

## 2. Identifying Senescent Brain Cells

Heterogeneity of senescent cells has been revealed through transcriptomic profiling [14,15,16]. The senescence phenotype is guided by context differences in cell type, upstream stressor and environment. To yield a deeper understanding of cellular senescence signatures across distinct cell types in vivo, a careful examination of multiple key biomarkers is needed and has been the subject of many reviews, for example [17]. Below we provide an overview of this strategy, with specific focus on its utility to identifying senescent post mitotic brain cells.

### 2.1. Absence of Proliferation/Stable Cell Cycle Arrest

Various indicators of cell cycle arrest such as cell cycle inhibition and telomere attrition have been discussed for mitotically competent brain cells, for example [18]. Despite the persistent dogma that neurons permanently withdraw from the cell cycle upon terminal differentiation, numerous studies have demonstrated the expression of cell cycle proteins in post-mitotic neurons, that can give rise to dysfunctional hyperploid cells [19]. A decrease in telomere length is associated with cell division; notably telomere shortening has been observed in non-replicative neural brain populations in C57BL/6 mice in a cell cycle-independent manner [20]. Telomere shortening may occur in replication-independent scenarios in long-lived post-mitotic cells through oxidative stress and downregulation of telomeric factor *POT1* and shelterin subunit *TRF2* [21]. While markers of proliferation may not necessarily indicate mitotic competency in post-mitotic cells, they may reflect cell cycle re-entry out of G0 into G1 which could make them vulnerable to apoptosis or senescence. Cell cycle re-entry has been estimated to occur in ~11.5% of post-mitotic cortical neurons through DNA content variation and ~20% of post-mitotic neurons in AD through both DNA content variation and expression of cyclin B1 [22,23]. The data collectively indicate a link between aberrant neuronal cell cycle activity and neuronal dysfunction and disease. For example, multiple studies link AD associated Ab [24,25,26] and phosphorylated tau with aberrant cell cycle activity [27,28,29]. Cell cycle re-entry in the absence of AD pathology also has been described [30] and put forth as a potential novel therapeutic target for neurodegenerative diseases [31]. In this context, an open question remains whether proliferation markers may also apply to a pre-senescent phase in post-mitotic neurons. Co-expression of cell cycle mediators in post-mitotic cells, such as G1 proteins, in the absence of apoptotic markers (i.e., caspase 3) suggests an arrest of aberrant cell cycle activity, consistent with senescence. Alternatively, post-mitotic quiescent cells may more readily transition to senescence than mitotically competent cells, for example through changes in lysosomal activity [32]. Nonetheless, measuring molecular signatures of cell cycle activity may provide evidence for or against a senescence stress response in post-mitotic cells. Examples of studies looking at stable cell cycle arrest are discussed in Section 3.

### 2.2. Cell Death Resistance

Senescent cells display enhanced survival over their non-senescent counterparts by activating senescent cell anti-apoptotic pathways (SCAPs) [33,34]. Similarly, post-mitotic cells, including neurons, acquire a greater resistance to cell death as they mature [35,36]. A complete understanding of neuronal cell death resistance is not known; however, some pathways have been identified [37,38,39,40]. Similarly, while some SCAPs have been identified for senescent cells, this is a burgeoning research area. It is tempting to speculate that SCAP-mediated degeneration resistance may contribute to (or use similar mechanisms as) post-mitotic cell death resistance. In this way, identifying molecular regulators of cell death resistance in post-mitotic cells may apply to senescence, and vice versa. In response to injury, mitotically competent cells may proliferate; instead post-mitotic cell cycle reentry triggers degenerative processes [41]. In this review, we provide evidence that post-mitotic senescence in neural tissue may preserve cellular integrity by avoiding cell death [42]. Thus, cell cycle inhibitors such as INK4 cyclin-dependent kinases inhibitors (CKIs) (i.e., p16, p18, p19) and CK-interacting protein/kinase inhibitor protein CKIs (i.e., p21, p27, p57) may be protective and contribute to cell death resistance in post-mitotic cells [41]. Though a stable senescence cell cycle arrest may confer degeneration resistance to the affected post-mitotic cell, downstream consequences of its preserved survival may include neural network dysregulation and chronic inflammation through secreted factors.

### 2.3. Secretory Phenotype

Post-mitotic cells can produce a senescence-associated secretory phenotype (SASP) consistent with that of mitotically competent cells. For details on the definition, role, and common factors used to identify SASP in the brain, please refer to previous reviews on this topic [18,43,44]. Briefly, upregulation of NFκb activity and consequential production of canonical pro-inflammatory markers may occur in post-mitotic cells similar to that of dividing cells. Moreover, post-mitotic cells may produce unique SASP factors, like aggregation-prone proteins that impact protein homeostasis and drive neurodegenerative proteinopathies. For example, human postmortem brains from patients clinically diagnosed with AD, PD, and dementia with Lewy bodies (DLB) all show severe hyperphosphorylated microtubule-associated tau, β-amyloid, and α-synuclein loads, although topographical distribution of protein aggregates are different [45]. These protein aggregates may influence each other and synergistically promote the accumulation of one another [46,47]. Thus, SASP is an important component of post-mitotic senescence with implications for neurodegeneration.

### 2.4. Senescence Associated β-Galactosidase

In-depth details of the SA β-gal assay specific to brain tissue were described in our previous review [18]. Briefly, SA β-gal detects lysosomal-β-galactosidase activity at pH 6.0 [48]. While useful for distinguishing senescent cells in culture, it is detected in brain tissue independently of age or senescence [49]. The *GLB1* gene encodes for lysosomal beta-D-galactosidase and is the origin of the SA β-gal activity, but SA β-gal’s cellular roles and mechanism in senescence are not fully understood [50]. In post-mitotic cells, interpretation of SA β-gal requires extra caution. Purkinje neurons in the cerebellum, CA2 neurons, and a subset of cortical neurons all display SA β-gal even in young mice [49]. A recent study demonstrated that lysosomal activity mediates the transition from deep quiescence to senescence [32]. Given that neurons are quiescent, assigning positive staining to senescence versus quiescence becomes especially challenging in static tissue. Even in vitro, SA β-gal positivity has been reported in neurons in the absence of other markers of senescence [51]. A more detailed review of SA β-gal used to identify senescence in post-mitotic cells is described in later sections.

### 2.5. Concluding Remarks on Identifying Senescent Cells

Evaluating senescence requires in-depth understanding of the cell type and markers of senescence. Post-mitotic cells have different characteristics from mitotically competent cells that should be considered when evaluating for senescence. In the following sections, we review studies reporting senescence in post-mitotic brain cells. We evaluate the methods used by key studies and compare them to our pre-defined criteria presented above and summarized in Figure 1.

## 3. Neurons

The adult human brain contains an estimated 86 billion neurons [52]. Barring neurodegenerative disease or brain trauma, nearly all cortical neurons (96–98%) remain alive during the lifespan. Their exceptional survival has been attributed to their restriction of apoptotic pathways, though the precise molecular details are not fully understood [16]. An appreciation for dysfunctional, not missing, neurons has emerged over the past decade. For example, age-associated cognitive decline has been attributed to changes in neuronal chemistry, metabolism, and/or morphology, but not necessarily the progressive loss of neurons [53]. Re-evaluation of the literature and accumulating experimental evidence suggests that age- and disease-induced stressors on neurons initiates a neuronal senescence stress response as a means to avoid active degeneration and cellular loss. However, as we discuss, these neuronal structural and functional changes contribute to pathogenesis in neurodegenerative diseases [49,54,55,56]. For example, p16 and p21 expression has been reported in neurons and glial cells in postmortem motor and frontal association cortex of ALS patients [57], while microglia express p16, p53, and SASP in late-stage spinal cord of the ALS rat model [58] (please refer to mitotic review for microglia senescence [18]). As terminally differentiated cells arrested in G0, neurons either inherently fulfill one of the key defining features of senescence (near permanent cell cycle arrest); alternatively, they may arrest in G1 after cell cycle re-entry, which has been described in numerous degenerative diseases [22,23] (Figure 1). This phenomenon is not unique to neurons; a recent review provides a discussion on the topic of post-mitotic senescence across tissues [42]. Here we review supporting evidence that neurons, like mitotically competent cells, have the ability to mount a canonical senescence stress response.

### 3.1. Neuronal Senescence in Tauopathies and Peripheral Neuropathies

Senescent cell heterogeneity, in part, arises from differences in cell biology of the parent cell. Growing experimental evidence demonstrates that the phenotypic diversity of neuronal senescence reflects the heterogeneity of neuronal subpopulations. Historically, neurons have been classified by morphology, anatomical location and/or distinct shapes, and function that can be further classified by direction, action on other neurons, discharge patterns, and neurotransmitter utilization. Recent methodologies, in particular single nucleus transcriptomics, have provided an even deeper insight into neuronal heterogeneity [59]. For example, MAPT encodes the microtubule-associated protein tau proteins, which are often referred to as “neuron-specific” or “axon-specific” proteins. However, the diversity in tau proteins arise from extensive processing at the mRNA and protein levels. Six major tau protein isoforms are expressed in the adult brain which arise through alternative splicing; post-translational modifications further amplify the tau protein diversity by producing dozens of unique forms of tau protein that are differentially regulated and expressed based on developmental age and neuronal subtype [60,61]. Tau protein accumulation is the most common intraneuronal pathology among neurodegenerative diseases, though neuropathology and clinical presentations differ across diseases [62]. Among tauopathies, neurons containing neurofibrillary tangle (NFT) aggregates of heavily post-translationally modified tau are the closest correlate with neurodegeneration and dementia in AD, yet they are long-lived [63]. We recently determined that these neurons display a canonical senescence stress response [49]. Analyzing transcriptomic data from postmortem human brain provided the opportunity for a within-subjects comparison between neurons with or without NFTs. The transcriptomic and pathway analyses revealed expression patterns in NFT-bearing neurons consistent with senescence including upregulated anti-apoptotic/pro-survival pathways and concomitant inflammatory and secretory pathways. Using four independent tau transgenic mouse models, we found evidence for DNA damage; aberrant cellular respiration; karyomegaly; upregulation of cell cycle inhibitors, inflammation and inflammatory mediator NFκB. These phenotypes occurred concomitant with NFT formation and were reduced by genetically removing endogenous tau protein, indicating a molecular link between tau and neuronal senescence. Moreover, intermittent treatment with senolytics (dasatinib plus quercetin) caused ~35% reduction in NFTs that coincided with a reduction in senescence-associated gene signature (cell cycle inhibitors and inflammation). We did not observe neuronal senescence phenotypes in thalamic, midbrain, or cerebellar neurons. It remains unknown whether or not high expression of transgenic tau could ultimately drive neuronal senescence in these other neuronal subpopulations. However, work from other groups suggest that midbrain [56] and cerebellar neurons [54] may utilize other molecular mediators aside from MAPT/tau.

Overexpression of human non-mutated tau and its persistent phosphorylation also contributes to peripheral neuropathy and memory deficits [64]. Long-term and short-term memory were significantly impaired in female transgenic mice expressing all six human tau isoforms [64]. Peripheral neuropathy was evidenced as motor nerve conduction velocity (MNCV) slowing, paw tactile allodynia, paw heat hypoalgesia, and low paw density of intraepidermal nerve fibers in human tau mice compared to wild type mice [64]. Notably, neuronal senescence has been associated with cisplatin-induced peripheral neuropathy (CIPN). Primary DRG neurons treated with cisplatin upregulate SA β-gal activity and expression of *Mmp*-9, *Cdkn1a, Cdkn2a*, and display elevated translocation of HMGB1 compared to controls [65]. In a mouse model of CIPN, dorsal root ganglia (DRG) neuronal populations upregulated the DNA damage response pathway and *Cdkn1a* gene expression as determined by single-cell RNA-sequencing. Neuronal senescence was further verified by increased protein expression of p21, p-H2AX, NFκB-p65; SA β-gal; and lipofuscin granules [66]. Clearing p16 and/or p21 positive cells either pharmacologically with ABT263 or by utilizing suicide gene therapy (i.e., p16-3MR ganciclovir/herpes simplex virus thymidine kinase system) reversed CIPN as evidenced by improved mechanical and thermal thresholds [65]. Collectively these studies indicate senescence-associated neuronal dysfunction in the central and peripheral nervous system where tau may be linked.

### 3.2. Neuronal Senescence in Parkinson’s Disease

The H1 MAPT haplotype [67] and single nucleotide polymorphisms in MAPT have been associated with age of onset and progression of PD [68]. Despite the strong genetic association with MAPT, tau pathology occurs only in about 50% of patients with PD [69]. Though cognitive dysfunction may occur, PD primarily affects motor behavior. Neurodegeneration in PD predominantly occurs in the substantia nigra where up to 70% of dopaminergic neurons can be lost in late disease stages [70,71]. These neurons express significantly lower levels of MAPT and tau protein than those in the cortex or hippocampus (~4-fold and ~6-fold difference, respectively) and do not develop tau pathology [72]. Instead, the hallmark protein deposit in PD is α-synuclein. Experiments in cell lines suggest that α-synuclein expression levels differentially regulate the cell cycle [73]; however, conclusive studies demonstrating α-synuclein-mediated neuronal senescence have not been reported. To date, the most comprehensive work on dopaminergic neuronal senescence involves special AT-rich sequence-binding protein 1 (SATB1) [56]. SATB1 functions as a transcription factor and chromatin architecture organizer [74]. SATB1 is overexpressed in various tumors and has been referred to as a T-cell-specific transcription factor given its importance in T cell development. A meta-analysis of genome-wide association studies comparing PD cases with controls identified SATB1 as a candidate risk gene [75]. Neurons in PD-vulnerable brain regions (e.g., substantia nigra pars compacta) display lower levels of SATB1 than neurons from the less susceptible ventral tegmental area [76]. Genetically reducing SATB1 in dopaminergic neurons drives a neuronal senescence response including elevated p21 protein expression, karyomegaly, SASP, and mitochondrial dysfunction [56]. SA β-gal and lipofuscin, hallmarks of mitotically competent senescence that also co-occur in neurons, were also observed. Mechanistically, SATB1 repressed dopaminergic neuron senescence by binding the regulatory region of CDKN1A. In the absence of SATB1, the CDKN1A encoded protein, p21, expression level increased to perpetuate the neuronal senescence stress response. When the authors reduced Cdkn1a in SATB1 knockout neurons, fewer senescent cells (as determined by SA β-gal) were observed, providing evidence for the mechanistic link between SATB1-p21 mediated neuronal senescence. Interestingly, reducing SATB1 in cortical neurons did not modulate Cdkn1a/p21 levels, which was attributed to a more open Cdkn1a locus in dopaminergic than cortical neurons. In contrast, tyrosine hydroxylase expressing neurons require Satb1 expression for their survival and will undergo neurodegeneration within three weeks of downregulated Satb1 [76]. This observation indicated that de-repression of Cdkn1a and concomitant increased p21 expression caused apoptosis and clearance by microglia. Evidence for this was observed by neuronal SASP production and concomitant microglia co-localization with tyrosine hydroxylase positive neurons. Follow-up experiments to deplete microglia after Satb1 reduction would conclusively demonstrate whether or not Cdkn1a-expressing dopaminergic neurons fulfill the criterion of apoptosis resistance in neuronal senescence. Indeed, elevated p21 expression induced apoptosis in vitro, indicating that it may not confer neuronal apoptosis resistance [77]. Nevertheless, the study by Riessland et al. determined a dopaminergic neuron-specific role of SATB1 in modulating Cdkn1a/p21 expression and downstream senescence phenotypes including karyomegaly, mitochondrial dysfunction, production of SASP, lysosomal dysfunction and presence of SA β-gal and lipofuscin [56].

In PD, disease-related stressors on neurons contribute to defects in several cellular systems ultimately involving alterations in Bcl-2 family signaling, JNK activation, p53 activation, expression of cell cycle regulators [78]. While many of these processes including those addressed above are thought to contribute to neuronal degeneration, some are also hypothesized to reflect survival-promoting mechanisms such as senescence. More recent studies focused on neuronal senescence in PD have revealed that overexpression of mutant p53, p21, or mutant Leucine-rich repeat kinase 2 (LRRK2) increased SA β-gal, and αSyn protein expression and fibril accumulation in vitro [77]. Transgenic mice expressing the same mutant LRRK2_G2019S_ displayed elevated oligomeric αSyn, β-galactosidase and p21 expression. The increase in αSyn was due to impaired degradation, not increased transcription [77]. The results suggest that the LRRK2_G2019S_ mutation may activate the p53-p21 senescence pathway, which is upstream of α-synuclein accumulation. While suggestive of senescence, the study did not evaluate cell cycle activity, apoptosis resistance or SASP production in the affected cells. Nonetheless, future studies to dissect if or how PD mutations may interact with *SATB1* would elucidate whether these pathways converge on a common senescence-associated pathway relevant to PD pathogenesis. An intellectual framework for proteinopathy-induced senescence in neurodegenerative diseases was first proposed in 2009 by Golde and Miller [79]. The idea warrants further studies as the emerging data from mechanistic studies that have directly tested this hypothesis (i.e., tauopathy, α-synucleinopathy, and β-amyloid) indicate that post-mitotic neurons are especially vulnerable to protein aggregation stress as highlighted in these aforementioned studies.

### 3.3. Neuronal Senescence in Aging

Senescent cells accumulate with advanced age even in the absence of disease. In 2012, Diana Jurk et al. evaluated neuronal senescence in naturally aged mice with or without increased DNA damage by genetically manipulating telomerase [54]. Age-associated DNA damage was associated with neuronal senescence in the brains of 32-month-old mice [54]. DNA damage foci, as determined by gH2A.X immunostaining, was elevated in cerebellar Purkinje and cortical neurons from 32-month-old mice compared to 4-month-old mice. These neurons also displayed activated p38 MAPK (phosphorylated at Thr^180^/Tyr^182^) indicative of DNA double strand breaks. Oxidative stress was assessed by visualizing cells with elevated lipid peroxidation product, 4-hydroxynonenal (4-HNE). Immunostaining with 4-HNE revealed cytoplasmic granular accumulation within the same subpopulations of cells. Similarly, these large neurons expressed higher levels of inflammatory protein IL-6 than other cell types. SA β-gal activity and lipofuscin (as measured by autofluorescence) showed similar overlapping patterns. Given the overlapping co-staining of multiple marker combinations, the authors hypothesized that DNA damage (gH2AX) increased with advanced age, which activated the DNA damage response (p-p38 MAPK) to induce a senescence-like pro-inflammatory (IL-6) and pro-oxidant phenotype (4-HNE) similar to mitotically competent cells (lipofuscin and SA β-gal). To begin evaluating mechanistic mediators of the senescence phenotype, they utilized transgenic mice with telomere dysfunction with or without *Cdkn1a*. Neurons from mice with telomere dysfunction (late generation telomerase knockout mice, F4 TERC-/-) displayed elevated levels of gH2AX, p-p38MAPK, 4HNE and IL6 compared to those with one functional copy of TERC. The genetic removal of *Cdkn1a* modulated these phenotypes in mice regardless of telomerase activity, however genotype and cell type specific phenotypes were observed. For example, in TERC wild type mice, the absence of p21 only significantly altered the 4HNE phenotype and only in Purkinje neurons (not cortical neurons). In contrast, in mice with telomere dysfunction, removing p21 did not modulate 4HNE. Instead, the absence of p21 significantly reduced gH2Ax and IL6 in Purkinje neurons and p-p38 and IL6 in cortical neurons. These results again highlight heterogeneity of the senescence stress response unique to different neuronal subpopulations. Nonetheless, removing p21 robustly reduced inflammation, as assessed through IL6, in both cellular populations to provide evidence that neuronal senescence may contribute to sterile inflammation with advanced age.

Insulin provides trophic support and drives excitatory signaling in neurons [80,81]. A loss of neuronal sensitivity to insulin, referred to as insulin resistance, coincides with their dysfunction and disease. The mechanisms driving insulin resistance in brain cells are not well understood, but risk factors include advanced age, obesity, peripheral insulin resistance, and metabolic dysfunction [82,83]. Recent studies in mice have demonstrated that brain insulin resistance induces neuronal senescence, which leads to synaptic dysfunction [55,84]. In these studies, insulin resistant neurons display several molecular, functional and morphological changes consistent with senescence [55]. Specifically, mice that developed spontaneous peripheral insulin resistance at either young (3-months-old) or old (24-months-old) age also displayed signs of brain insulin resistance (i.e., elevated insulin in the CSF, elevated pIRS1 (Ser^307^ and Ser^612^)], and senescence (i.e., neurite loss, elevated Cdkn1a and Cdkn2a and SA β-gal activity). This finding indicates that insulin resistance, like tau accumulation or loss of *SATB1,* may drive premature neuronal senescence in the absence of advanced age. The insulin resistant mice, regardless of age, behaved poorly on cognitive behavior tasks to indicate that neuronal insulin resistance/senescence co-occurred with poor brain function. Mechanistically, chronic insulin was shown to reduce hexokinase 2, impair glycolysis and increase levels of p25, a potent activator of both CDK5 and GSK3β. The simultaneous signals from CDK5 (neuronal cell death) and β-catenin (cell cycle re-entry) pushed neurons to enter a senescence-like state. A detailed signal transduction cascade was elucidated in vitro whereby insulin increased Ccnd1 and Cdkn2a expression, nuclear localization of β-catenin, cyclin D1 and p19ARF. The increase in p16INK4a and PML occurred later. Aberrant β-catenin also induced a parallel p53-p21 senescence pathway. The authors concluded that chronic insulin signaling induced a neuronal senescence phenotype through the over-stabilization and nuclear localization of β-catenin. Tau phosphorylation was not assessed, but given the increased activity of tau kinases Cdk5 and GSK3β and parallels with findings in Musi et al. [49], it is tempting to speculate that aberrant tau may also contribute to insulin resistance-mediated neuronal senescence.

### 3.4. General Considerations for Evaluating Neuronal Senescence

Observations across the aforementioned studies highlight the complexity of applying canonical senescence measures to post-mitotic cells. For example, we caution the use of lipofuscin and SA β-gal for neuronal senescence as these markers seemingly reflect shared phenotypes among neurons, across age and/or disease, that requires further investigation into their association with other senescence markers. The best example are cerebellar Purkinje neurons that display SA β-gal throughout the lifespan [49]. Jurk et al. noted, “the frequencies of neurons showing multiple markers of a senescent phenotype are very substantial, going well beyond 20% in Purkinje cells already in young mice brains” [54]. A key readout for this conclusion was SA β-gal staining. Given the early stages of defining neuronal senescence in vivo, it remains unknown whether SA β-gal positivity truly reflects senescence in Purkinje neurons, which could become senescent in early life due to their high energetic and metabolic demands. Other studies have demonstrated that cerebellar Purkinje neurons can survive and function as polypoid cells [85]. Neuronal polyploidy suggests that DNA replication occurred, but that neuronal mitosis stalled. Indeed, hyperploid neurons have been reported in preclinical and mild stages of AD as evidenced by immunofluorescence and slide-based cytometry methods cross-validated by chromogenic in situ hybridization [86]. The neurons avoid apoptosis, upregulate several cell cycle mediators and survive months in the adult mouse brain, which meets several criteria of a senescent cell. Importantly, cerebellar Purkinje neurons are indispensable for motor movement control. Notably, gait speed, coordination, and balance are significant predictors of mortality [87,88]. It is tempting to speculate that senescence of these neurons may contribute to the overall decline in health and increased mortality with advanced age. Alternatively, the physiological function of these neurons may require signaling through cellular and molecular pathways resulting in phenotypes typically attributed to senescence. For example, we routinely observe neuronal lipofuscin throughout the lifespan, though it notably increases with age; similarly, we observed high levels of SA β-gal activity in these same neuronal populations throughout the mouse lifespan [49]. Moreno-Blas et al. also proposed that SA β-gal may not be a reliable marker of senescence by itself [89]. Despite cortical neurons expressing senescence-associated phenotypes such as p21, γH2AX, ruptures of DNA, lipofuscin, SASP, and irregular nuclear morphology, they observed normal nuclear morophology in some neurons with high SA β-gal [89]. Instead, their data suggested that autophagy impairment/dysfunction, perhaps through lysosomal fusion with autophagosomes, critically contributed to the neuronal transition from quiescence to senescence, similar to that reported by [32]. Since SA β-gal positivity overlaps with lysosomal dysfunction, it may be useful to narrow down potential senescent cell candidates; however, as indicated by Moreno-Blas [89] and several studies reviewed here (i.e., [49,51,54]) it cannot be used in isolation. Similarly, neuronal lipofuscin staining was first reported in children in 1903 and has been later confirmed in several studies where it occurs in at least 20% of neurons by 9-years-old [90]. These aggregates of oxidation products of lipids, proteins, and metals autofluoresce non-specifically bind antibodies which can complicate interpretations of immunofluorescence assays and thus requires multiple controls. The pigment granules change with aging by increasing progressively in size, as well as their subcellular localization thus appropriate age-matched negative controls and antibody controls are necessary to interpret results. Within the aging field, the increased rate of lipofuscin formation and accumulation is considered a hallmark of both replicative and stress-induced senescence [91,92] and methods for its specific staining (i.e., Sudan Black B) are increasingly used to detect senescence in vitro and in vivo [93]. It is our opinion that at this time both lipofuscin and SA β-gal require further investigation before using them as decisive markers for neuronal senescence.

### 3.5. Concluding Remarks

Differentiated neurons are remarkably apoptosis resistant, but their vulnerability to excitotoxicity increases with age [94]. Neurons inherently lack the option to divide, but they upregulate cell cycle proteins in response to stress. The inability to replace these critical cells indispensable for maintaining life may have placed strong evolutionary pressure to favor stress-induced senescence over apoptosis. In this way, neuronal survival would be maintained though the number of dysfunctional cells would increase with advancing age. Indeed, this is what is observed in the human brain [52,53]. As the burgeoning field of neuronal senescence advances, we expect that the next wave of studies will reveal additional molecular regulators, clarify pathways previously identified, and differentiate between shared pathways and neuron subtype specific mechanisms. Additionally, with the increasing use of single cell technologies, we anticipate an increased ability to identify, track and study senescence with greater clarity on the phenotype(s) and how they change across the lifespan and in disease.

## 4. Astrocytes

Astrocytes are an abundant and heterogenous cell population within the central nervous system (CNS). They comprise 20–40% of the total glial cell population in the brain, depending on region, developmental stage, and species [95,96,97]. Along with oligodendrocytes, astrocytes originate from the neural tube [98]. Astrocytes differentiate from the glial progenitor cells proliferating in the forebrain subventricular zone as they migrate outwards to other regions of the brain [99]. The majority of astrocytes are considered post-mitotic, and in the absence of pathology or disease, they display low rates of turnover and proliferation [100].

Astrocytes vary in function and morphology. Distinct types, including radial astrocytes, fibrous astrocytes, and protoplasmic astrocytes have been elucidated within the CNS based on structure, distribution, and function, as well as their expression level of the different isoforms and splice variants of the intermediate filament protein glial fibrillary acidic protein (GFAP) [101,102]. Astrocytes have been implicated in maintaining water and ionic homeostasis, providing metabolic and structural support to neurons, and regulating the blood–brain barrier (BBB) [102,103]. They also cooperate with microglia to control local neuroinflammation and neuronal restoration following damage to the CNS. Similar to microglia, astrocytes prune synapses and remove cellular debris within the synapse in healthy and diseased brains [104,105]. Genes crucial for astrocyte function such as Excitatory Amino Acid Transporters 1 (*EAAT1*) and 2 (*EAAT2*), potassium transporter *Kir4*, and water transporter *AQP4* involved in glutamate, glutamine, potassium, and water homeostasis in the brain have shown to be downregulated when astrocytes become senescent [106]. Thus, their change in function associated with senescence can lead to detrimental effects including the onset of various neurodegenerative pathologies [103,107,108,109,110].

Astrocyte senescence is often wrongly conflated with astrogliosis or astrocyte reactivity. Reactive astrogliosis involves structural changes to the astrocytes alongside cellular proliferation and migration [100,109]. Reactive astrocytes, also known as A1 cells, have been shown to be induced by activated neuroinflammatory microglia through the secretion of Il-1α, TNFα, and C1q cytokines. Upregulated expression of GFAP is a known marker of reactive astrocytes, and its levels are also increased during aging [111,112]. In contrast, radiation-induced senescent astrocytes demonstrated a downregulation of GFAP [113]. Reactive A1s lose their ability to promote neuronal survival, outgrowth, synaptogenesis, and phagocytosis and induce death of neurons and oligodendrocytes [114]. A1s have also been shown to be present in the brains in many neurodegenerative disorders, including AD, PD, and Huntington’s disease [114,115]. The benefit of astrogliosis and subsequent scar formation is the protection of the surrounding neurons and tissue and restriction of inflammation and pathology. However, dysfunction in reactive astrocytes can lead to neuronal dysfunction, and eventually degeneration that can contribute to various CNS disorders. Many of these features are similar with a senescence phenotype, including morphology changes and secretion of pro-inflammatory molecules.

Astrocytes undergo a senescence-like stress response, which has been referred to as “astrosenescence” and described as a functional change from neurosupportive to neuroinflammatory [116]. Oxidative stress, exhaustive replication, inhibition of proteasomes, and an increase in glucose concentration elicit an astrocyte response consistent with senescence, in vivo and in vitro (reviewed: [116,117]). For example, replicative senescent primary human fetal [118] and rat [119] astrocytes displayed an arrest of growth and cell cycle progression; the human fetal astrocytes also upregulated gene expression of *TP53* and *CDKN1A*. Astrocytes do not express TERT [120] and replicative senescence was not avoided with telomerase reverse transcriptase (hTERT) expression [118], indicating that telomere-length independent mechanisms govern replicative senescence in astrocytes. Inhibiting p53 function with human papillomavirus type 16 E6, however, delayed the onset of senescence, implying a p53-dependent mechanism of replicative senescence in astrocytes [118]. Increased SA β-Gal activity, marked by staining kits, was also observed in many of these studies [113,117,121,122]. Strengths and weaknesses of using this method for labeling brain cells have been discussed [18].

Radiation cancer therapy has potential to induce senescence [123]. The effect of radiation therapy on astrocytes in vivo was examined by evaluating human brain from individuals receiving cranial radiation cancer therapy [113]. Senescent cells were identified with immunohistochemical labeling of p16, heterochromatin protein Hp1γ, and expression of Δ133p53, an inhibitory isoform of p53. Elevated p16 and Hp1γ largely co-localized with astrocytes in patient brains that had received radiation, but not in control tissue. Expression of Δ133p53 was primarily in astrocytes, and its role in senescence was explored in vitro. They found that these irradiated astrocytes in vitro had diminished Δ133p53, and developed a phenotype associated with other senescent cells, such as increased SA β-Gal activity, p16, and IL-6. However, restoration of Δ133p53 expression inhibited and prevented further senescence, promoted DNA repair, and prevented astrocyte-mediated neuroinflammation and neurotoxicity [113]. Collectively, this study [113] and others [106,124] have characterized the radiation-induced senescence phenotype in astrocytes to include decreased proliferation and increased SA β-Gal activity, along with typical increased expression of p53, p21, and p16, which were analyzed using Western Blot [113,121].

Senescent astrocytes downregulate genes associated with activation, including GFAP and genes involved in the processing and presentation of antigens by major histocompatibility complex class II proteins, while upregulating pro-inflammatory genes [121]. Increased expression of p16, p21, p53, and MMP3 have also been associated with astrocytes undergoing senescence and those isolated from aged brains [125]. The downregulation of genes associated with development and differentiation, coinciding with the upregulation of pro-inflammatory genes, manifest as functional changes (i.e., inflammatory stress response). This may perpetuate a pro-inflammatory feedback loop that is stably maintained by senescence-associated changes in gene expression and transcript processing [126]. 

Astrocyte senescence increases with age in the human brain and in AD [127,128] and PD [129]. The consequences of astrocyte senescence are myriad. Functionally, astrocytes communicate with nearby neurons and the surrounding vasculature to clear disease-specific protein aggregates, including β-amyloid, the accumulation of which has been linked to the progression of AD [96,130]. The release of SASP factors by senescent astrocytes including IL-6, IL-8, MMP3, MMP10, and TIM2 were found to contribute to β-amyloid accumulation, phosphorylation of tau protein, and an increase in NFTs [125,131]. An increased risk for PD has been linked to contact with the herbicide paraquat (PQ), which an environmental neurotoxin. Complementary in vivo and in vitro approaches were used to evaluate mouse and human astrocyte responses to PQ [129]. PQ-treated astrocytes developed several features consistent with senescence, including upregulated *Cdkn2a/*p16. Importantly, senescent cell removal improved neurogenesis in the subventricular zone, reduced neuronal loss and rescued motor function deficits in PQ-treated mice [129]. Collectively their results highlight astrocyte senescence as a mechanism of PQ-associated neuropathology and brain dysfunction, and represents an appealing therapeutic target for the treatment of PD.

### Concluding Remarks

“Astrosenescence” is a complex and heterogeneous process that necessitates evaluating astrocyte structure, distribution, function and molecular expression profiles. Measuring the expression level of GFAP [113,125] can help differentiate whether upregulated pro-inflammatory cytokines and chemokines expression reflect astrogliosis or astrosenescence [125,131]. The most consistently shared features across senescent astrocytes were arrest of growth and cell cycle progression, increased expression of p53 and p21, and p16 [113,117,121] and some evidence of increased SA β-Gal activity [117]. Collectively the studies reviewed here indicated that functional changes associated with senescent astrocytes contribute to chronic neurodegenerative diseases and may propagate inflammation and induce senescence to surrounding cells [132,133]. Targeting them for removal represents an opportunity to intervene in neurodegenerative diseases.

## 5. Endothelial Cells

Endothelial cells form a single layer of cells called endothelium that line the blood vessels of the circulatory system. They have an array of functions in vascular homeostasis such as regulating blood flow, immune cell recruitment, maintaining blood vessel tone, and hormone trafficking [134,135]. While endothelial cell function is heterogeneous and tissue-specific, several studies have demonstrated that endothelial cells can become senescent in adipose tissue, coronary arteries, and in the human umbilical cord using observations of morphology changes, SA β-gal activity, and SASP through DNA microarray [136,137,138]. While there is a great literature describing senescent endothelial cells throughout the body, the focus of this section turns to brain microvascular endothelial cells.

Brain endothelial cells are mostly post-mitotic with minimal proliferation [139,140,141]. They express a high density of tight junction and adherens junction proteins and high transendothelial electrical resistance [142,143,144]. Functionally, brain endothelial cells contribute to the BBB, regulate local cerebral blood flow as a part of the neurovascular unit (NVU), and thus have important implications for brain diseases [145,146,147]. The BBB is a highly selective semipermeable barrier with tight junctions that closely regulates the biochemical composition of the brain by restricting the free diffusion of nutrients, hormones, and pharmaceuticals [148]. The tight junctions force molecular traffic to take place through the endothelial membrane through sealing of the paracellular space and by establishing a polarized, transporting epithelial and endothelial phenotype [149]. During aging, endothelial cells experience senescence-associated stressors including oxidative stress, DNA damage accumulation, telomere shortening, increased NFκB signaling and decreased *Sirt1* expression [136,150]. Recent studies suggest that brain endothelial cell senescence could contribute to BBB dysfunction though neurovascular uncoupling and reactive oxygen species [151]. Indeed, increased BBB permeability and vascular dysregulation have been observed in patients with early cognitive dysfunction, cerebral microvascular diseases, and AD [152,153,154]. However, the co-occurrence of senescent endothelial cells with aging and disease makes it difficult to discern whether they are upstream mediators or downstream consequences of diseases. BBB dysfunction has been observed in patients with AD, Multiple Sclerosis (MS), traumatic brain injury (TBI), and stroke, featuring overexpression of MMP-2 and MMP-9 [155,156,157,158,159]. Molecular cascades such as activation of MMP’s have been suggested to induce senescence [160]. Thus, it is possible that brain insult leading to BBB dysfunction causes senescence as well.

There have been several recent studies specifically examining cerebrovascular endothelial senescent cells induced by external stimuli and natural aging. In one study, rat primary cerebromicrovascular endothelial cells were delivered 2–8 Gy of γ-irradiation using a 137Cs gamma irradiator [161]. After irradiation, cerebromicrovascular endothelial cells’ DNA damage was examined using the Comet Assay with alkaline single-cell gel electrophoresis followed by fluorescent imaging of the nuclei. Comet Assay visualizes the amount of DNA which leaves the nucleus as a marker for DNA strand breaks [162]. Irradiation caused DNA fragments to migrate out of the nuclei, indicating increased DNA damage [161]. SA β-gal staining was positive in a dose-dependent manner, p16 and p53 upregulation was observed, and a SASP was seen with upregulation of *IL-6*, *IL-1*α, *GM-CSF*, *G-CSF*, *MIP-1α*, *MCP-1*, eotaxin, and *IL-1β* via RT-PCR to verify senescence [161]. In another study discerning cerebromicrovascular endothelial cell senescence, 3-month-old and 28-month-old C57BL/6 mice gene expression profiles were compared by single-cell RNA sequencing [163]. The mean expression of senescence core genes (*Cdkn2a*, *Bmi1*, *Trp53*, *Hmga1*, *Chek1*, *Chek2*, *Prodh*, *Tnfrsf10b*, *Cdkn1a*, *Dao*), senescence effector genes (*Ppp1ca*, *Ahcy*, *Brf1*, *Map2k3*, *Map2k6*, *Smurf2*, *Tgfblil*, *Srsf1*, *Angptl2*), and SASP genes (*Ccl2*, *Ccl24*, *Ccl3*, *Ccl5*, *Ctnnb1*, *Cxcl1*, *Cxcl10*, *Cxcl12*, *Cxcl2*, *Cxcl16*, *Hgf*, *Hmgb1*, *Icam1*, *Igfbp2*, *Igfbp3*, *Igfbp4*, *Igfbp6*, *Igfbp7*, *Il15*, *Il18*, *Il1α*, *Il1β*, *Il2*, *Il6*, *Mif*, *Mmp12*, *Mmp13*, *Mmp14*, *Pgf*, *Plat*, *Timp2*, *Serpine1*, *Ccl3*, *Ccl4*, *Ang*, *Csf2*, *Kitl*, *Serpine2*, *Tnfrsf1a*, *Hgfi*, *Nrg1*, *Ereg*, *Areg*) were used to calculate a running enrichment score [163]. Higher senescence gene enrichment scores were found in brains from aged versus young mice [163]. These SASP factors have potential to promote neuroinflammation and affect BBB integrity [164]. Future studies may further clarify the downstream impact of senescent endothelial cells on their neighboring environment [165].

Other studies also highlight the difficulty of disentangling cause and effect of brain cell senescence, aging and disease. Emerging evidence suggests aberrant angiogenesis, and potentially endothelial senescence, may occur as bystander effects of other cell’s SASP. Studies using the rTg4510 mouse model of tauopathy have revealed an increased number of blood vessels and concomitant upregulation of angiogenesis-related genes such as *Vegfa*, *Serpine1*, and *Plau* [166]. Confocal imaging demonstrated aberrant vasculature near neurons with tau-containing NFTs which display a senescence-like phenotype (please refer to Section 3: Neurons) [49]. Together, these studies suggest that factors secreted by senescent NFT-containing neurons may negatively impact surrounding cells, which could drive aberrant angiogenesis. Alternatively, aberrant cerebrovasculature could be upstream of tau accumulation and contribute to NFT formation. To translate these studies to human clinical conditions, postmortem human AD brains with tau pathology were investigated for cerebrovascular senescence [167]. Cerebral microvessels were isolated from 16 subjects with a Braak NFT score of V/VI (B3) and 12 subjects with a Braak NFT score of 0/I/II (e.g., high neuropathology versus low neuropathology). Upregulation of senescence was inferred by elevated expression of *Serpine1*, *Cxcl8*, *Cxcl1*, *Cxcl2*, *Csf2*, and *Cdkn1a*; however, other markers of senescence were not evaluated [167]. Whether tauopathy causes endothelial senescence and induces a leaky BBB and/or endothelial senescence affects the vascular microenvironment will require further investigation [168].

### Concluding Remarks

Most of the aforementioned studies examined brain endothelial cell senescence by analyzing expression of senescence-associated genes [161,163,167]. Some studies also examined SASP genes [163,167]. Future studies are needed to evaluate cell cycle arrest, SCAPs, DNA-damage responses, resistance to apoptosis to define and validate senescence in brain endothelial cells [169,170]. Of interest for future studies will be determining brain region-specific differences in endothelial senescence and to better identify their mechanistic impact on the neighboring cells and environment.

## 6. Oligodendrocytes

Oligodendrocytes (OLs) are derived from oligodendrocyte precursor cells (OPCs) in a highly regulated process [171]. OPCs differentiate into pre-OLs, and later into mature, myelinating OLs in the presence of differentiation-promoting transcription factors [172]. The primary role of mature OLs includes myelination of neuronal axons in the CNS. Additionally, OLs play a role in providing metabolic support to myelinated axons, especially in axons that spike at high frequencies [173]. OLs have also been implicated in information processing, and defects in OL maturation are linked with behavioral abnormalities [173,174]. OLs are highly vulnerable to oxidative stress and mitochondrial injury, and OL loss occurs upon exposure to inflammatory cytokines [171,175,176]. OLs are also highly susceptible to accumulation of DNA damage during normal aging and have been indicated as a potential upstream cause of cellular aging leading to neurodegeneration, illustrated by the involvement of myelin in several neurodegenerative disorders [175,176]. DNA damage is a known mediator of senescence suggesting a potential relationship between senescence of oligodendrocytes and neurodegenerative disorders. Senescent OLs could result in defective myelination as seen in several neurodegenerative disorders [177,178,179]. For instance, loss of OLs can lead to demyelination as seen in MS.

Only a few studies exist that try to validate senescence in OLs. A rodent model with a novel senescence marker utilizing the p16 promoter, ZsGreen, crossed with the established APP/PS1 AD model was used to look for senescence in different cell types, including OLs [180]. OPCs showed upregulation of p21, p16, and SA-β-gal activity [180]. However, no senescence was observed in OLs (immunohistochemically stained with OL marker, CNP, and ZsGreen p16 senescence reporter) while OPCs were senescent and unable to differentiate into OLs [180]. It is likely that increased susceptibility of OPCs to the microenvironment increase the incidence of senescence in these cells compared to OLs. It may be possible that senescence in OLs occurs through p16-independent mechanisms. For example, a recent study reported an age-associated increase in p16-positive oligodendrocytes, but they were not cleared using senolytic approaches [181]. Brain cell type specific responses to senolytic clearance [181] highlights the heterogeneity of senescent cells even when in the same tissue, which may (in part) reflect cell type diversity in complex tissues such as the brain [59,182,183].

In a study of white matter lesions of frozen postmortem human brain tissue from patients who were over 65 years old, OLs exhibited elevated SA-β-gal [184]. Immunohistochemistry was used to double-label white matter tissue with SA-β-gal to identify cell types [i.e., astrocytes (GFAP+), microglia (CD68+), and oligodendrocytes (OSP+)]. Additionally, OLs also showed increased levels of 8-OHdG, a marker for oxidative stress, but did not display high levels of p16 [184]. Comparison of mRNA using qRT-PCR revealed a 1.5-fold increase in *TP53, H2AX,* and *CDKN1B* [184]. *CDKN1B* encodes p27kip1 and its upregulation results in the induction of a senescent phenotype [185]. Elevated *H2AX* and *TP53* are indicative of increased DNA damage and are also suggestive of a senescent phenotype. However, to confirm true senescence in these OLs, additional inspection of (1) SASP factors, (2) resistance to apoptosis through SCAPs, and (3) the presence of proliferation markers would be beneficial.

### Concluding Remarks

Although there are several studies that examine OPC senescence (refer to our other review on senescence in mitotically active brain cells for literature regarding OPC senescence) in multiple disease processes, there are limited data regarding the senescence of OLs in natural aging and other disease models. Limited studies mentioned suggest that SA-β-gal and gene expression analysis may be used to see if OLs have a senescent, but the results are inconsistent. For example, SA-β-gal activity was seen while high levels of p16 was not observed. Further study is required to establish the true senescence status of these cells and their potential role in aging and disease.

## 7. Summary

Cellular senescence has been best studied and characterized in mitotically competent cells [180]. However, most cells in the brain including neurons, astrocytes, endothelial cells, and oligodendrocytes have very low or no cell turnovers and show mostly post-mitotic phenotypes. Most post-mitotic brain cells that survive brain development will remain throughout the lifespan. While this feature historically precluded the study of senescence in the brain due to the early definition requiring mitotic competency, brain cell types are highly susceptible to acquiring a lifetime worth of damage known to drive senescence (Table 1). These include oxidative stress, DNA damage, and protein accumulation, which impact cell cycle and secretory phenotypes. While senescent cells continue to survive due to their apoptosis resistance, they tend to partially lose (or change) their function and increase expression of pro-inflammatory molecules. SASP from senescent cells can affect the microenvironment in the brain by its paracrine effect, causing other neighboring cells regardless of cell types to go senescent [133]. In the brain, these dysregulations manifest as an increase in neuroinflammation, increased BBB permeability, loss of neuronal synapses, demyelination, and dysregulated metabolism [179,186]. Collectively, these features have been associated with impaired cognition, and clearance of senescent cells as a therapeutic strategy has shown to reduce pathology, inflammation, and neuronal dysfunction [49,65,181,187].

Characteristics needed to positively identify senescent cells that were described in our previous review [18] can be translated from mitotically competent cells to post-mitotic cells, but with caution. For example, absence of proliferation using proliferative markers (i.e., BrdU, Ki-67) should rarely be considered to characterize post-mitotic senescence as most post-mitotic cells show a lack of proliferation and turnover. Instead, presence of these markers may indicate aberrant cell cycle activity consistent with either a fate of apoptosis or, potentially, pre-senescence. In neurons, SA β-gal staining was positive in even young Purkinje neurons. SA β-gal activity may be observed due to metabolic demands rather than as a marker for senescence. Astrocytes present their own challenge as reactive astrogliosis show senescence-like phenotype with changes in cell shape and secretion of pro-inflammatory molecules. Endothelial cells were evaluated as senescent mainly with gene expression analysis of senescence-associated genes, and there is a lack of verification with DNA damage, cell cycle arrest machinery, and resistance to apoptosis. OLs also showed elevated SA-β-gal activity but no activation of p16, which also questions the validity of the SA-β-gal assay in the brain. Although there is a lack of studies evaluating senescence in OLs, those reviewed here overall did not utilize all of our predefined criteria, similar to studies on dividing cells mentioned in our previous review [18]. Post-mitotic cells discussed in this review highlight the need to critically look at multiple markers of senescence.

Although senescence was initially, and exclusively, studied in mitotic cells, the literature reviewed here (Table 1) provides evidence that post-mitotic cells also undergo senescence as a complex stress response. The emergence of this new field, senescence in the brain, requires clarity of defining features. While tempting to label cells as “senescent” in many of these studies, a thorough evaluation of cell biology placed in the context of the cellular environment must also be considered. As a result, the field of neuroscience has pressured senescence biologists to clarify definitions and labels. As neuroscientists, we are in the early stages of applying methodologies and principles form senescence biology to the brain. In return, neuroscientists have over 60 years of lessons and principles of exceptional resistance to cell death to share with senescence biologists. A closer interaction and sharing of concepts between neuroscience and senescent cell biologists will propel both fields. As these efforts progress, we will continue to clarify definitions and revisit interpretations from the foundational studies reviewed here.

## Figures and Tables

**Figure 1 life-11-00229-f001:**
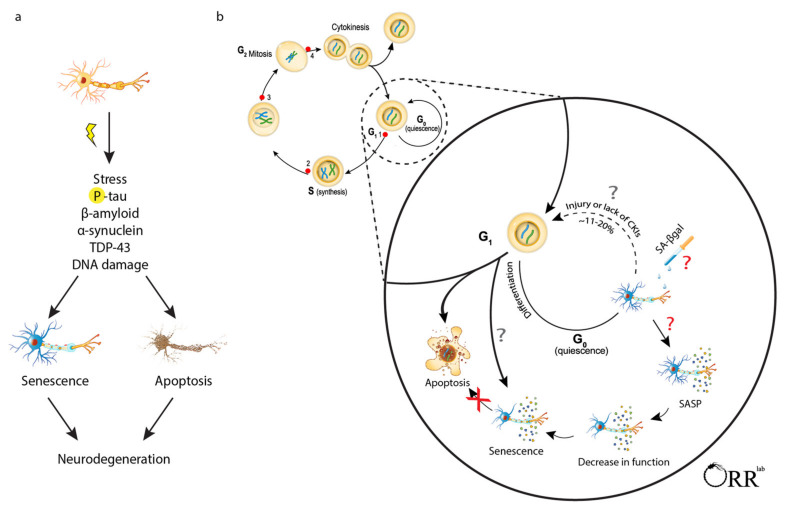
Senescence-associated mechanisms may confer exceptional resistance to cell death, but contribute to pathogenesis through inflammatory SASP. (**a**) The accumulation of stress, including protein aggregates and DNA damage, contributes to both senescence and apoptosis in post-mitotic cells. Protein aggregates including hyperphosphorylated tau, β-amyloid, and α-synuclein are seen in senescent cells, apoptotic cells, and patients with AD, PD, ALS and DLB. (**b**) Mitotic cells exit cell cycle and terminally differentiate into post-mitotic brain cells. Post-mitotic cell cycle re-entry can lead to cell death or senescence. In senescence, post-mitotic cells show stable cell cycle arrest, upregulate SCAPs, and release SASP. Neuronal SASP includes proinflammatory molecules and neurotoxic proteins. The SASP activates glia, drives inflammation, loss of neuronal connectivity, and perpetuates toxicity in a prion-like spread. SA β-gal can detect senescence, but it has a limited ability to discern between quiescent and senescent cells. AD: Alzheimer’s disease; PD: Parkinson’s disease; ALS: amyotrophic lateral sclerosis; DLB: dementia with Lewy bodies, CKI: cyclin-dependent kinases inhibitors; SCAPs: senescent cell anti-apoptotic pathways; SASP: senescence associated secretory phenotype.

**Table 1 life-11-00229-t001:** Biomarkers previously described to verify senescent cells.

Cell Type	Biomarkers	Reference
Neurons	SA-β-gal	[54], [55], [56], [65], [66], [77], [84], [89]
	H2ax	[49], [66], [89]
	Comet Assay	[89]
	Nuclear morphology	[89]
	Cytosolic HMGB1	[65]
	Telomere-associated DNA damage foci	[84]
	p53	[77]
	p21	[49], [54], [55], [56], [57], [65], [66], [77], [89]
	p16	[49], [55], [65], [84]
	SASP	[49], [54], [56], [65], [66], [89]
	Lipofuscin	[54], [56], [66], [89]
	p38MAPK	[54]
Astrocytes	SA-β-gal	[106], [113], [118], [119], [121], [122], [124], [125], [126], [127], [129]
	Heterochromatin protein Hp1γ	[113]
	Senescence-associated heterochromatin foci	[121]
	BrdU	[118], [129]
	53BP1 foci	[129]
	Lamins	[106], [122], [129]
	GADD45A	[122]
	p53	[118], [121], [122], [124], [125]
	p21	[118], [121], [122], [124], [125]
	p16	[106], [113], [118], [121], [126], [127], [129]
	SASP	[106], [113], [124], [125], [126], [127], [129]
	p38MAPK	[127]
Brain Endothelial Cells	Senescence-associated genes	[161], [163], [167]
	Comet Assay	[161]
	SASP	[161], [167]
Oligodendrocytes	SA-β-gal	[184]

## Data Availability

Not applicable.

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
