# Peer review of "The Cellular Senescence Stress Response in Post-Mitotic Brain Cells: Cell Survival at the Expense of Tissue Degeneration"

_life, 2021, doi:10.3390/life11030229_

Round 1

Reviewer 1 Report

This is a timely review that summaries the evidence supporting the significant impact of cellular senescence on neurodegeneration. It is unique by focusing on post-mitotic cells of the brain. It highlighted the essential characteristics of cellular senescence as multifaceted stress responses instead of merely a stable replication arrest. The information gathered is comprehensive and objective, serving as an excellent reference for further studying the pathogenesis of neurodegenerative disorders and exploring new avenues for therapeutic intervention. Publication is well-justified.

This reviewer suggests a few minor modifications to enhance the clarity:

  1. Given the complex and context-dependent nature of cellular senescence as well as the heterogeneity of brain cells, there may not be a standard set of markers to identify senescent brain cells. It might be good to point out the multifaceted nature of senescence phenotypes, and to emphasize that senescence should be determined by multiple key features, such as SASP, apoptotic resistance, proteostasis impairments etc. 
  2. There is, so far, no solid evidence to support mitosis, DNA replication, aneuploidy, or polyploidy in post-mitotic neurons in vivo. The experimental methods used to suggest these were indirect. It is better not to include these in this review, since it doesn’t enhance the main point, i.e. senescence can occur in post-mitotic cells.
  3. It doesn’t seem to be clear on how “astrosenescence” is conceptually distinguished from reactive astrocytes in the brain. Is it determined by GFAP levels?
  4. Although cellular senescence can occur in various cell types in the brain, given the paracrine effect of SASP, it may be helpful to mention that senescent cells can affect neighboring cells regardless of cell types in the brain in the summary section.
  5. It is impressive that the authors included a very comprehensive list of literature citation, but some key references were reviews. Citing additional original studies will strengthen this review article. 

Author Response

“1. Given the complex and context-dependent nature of cellular senescence as well as the heterogeneity of brain cells, there may not be a standard set of markers to identify senescent brain cells. It might be good to point out the multifaceted nature of senescence phenotypes, and to emphasize that senescence should be determined by multiple key features, such as SASP, apoptotic resistance, proteostasis impairments etc.”

Author response: This is an excellent point by the reviewer. Senescent cell heterogeneity is an important consideration when identifying senescent cells as described in line 96, 238, 386. We have further emphasized this statement in line 117 and 722. Line 119 guides the readers to examine the multiple key features of senescent cells that we describe throughout the manuscript.

“2. There is, so far, no solid evidence to support mitosis, DNA replication, aneuploidy, or polyploidy in post-mitotic neurons in vivo. The experimental methods used to suggest these were indirect. It is better not to include these in this review, since it doesn’t enhance the main point, i.e. senescence can occur in post-mitotic cells.”

Author response: We have added more references to support how aberrant neuronal cell cycle re-entry as well as polyploidy have been observed in vivo in lines 128, 141, and 433. We emphasize cell cycle re-entry specifically in the AD pathology as well as in the absence of AD pathology.

“3. It doesn’t seem to be clear on how “astrosenescence” is conceptually distinguished from reactive astrocytes in the brain. Is it determined by GFAP levels?”

Author response: Reactive astrocytes and senescent astrocytes share the feature of SASP. However, downregulation of GFAP and different gene expression patterns were observed in senescent astrocytes. In addition to the biomarkers that we summarized in line 585, we have added line 512 and 582 to better clarify this point.

“4. Although cellular senescence can occur in various cell types in the brain, given the paracrine effect of SASP, it may be helpful to mention that senescent cells can affect neighboring cells regardless of cell types in the brain in the summary section.”

Author response: Thank you, in addition to highlighting this in specific sections, we now have added a sentence to emphasize the effect of SASP on neighboring cells in line 757.

“5. It is impressive that the authors included a very comprehensive list of literature citation, but some key references were reviews. Citing additional original studies will strengthen this review article.”

Author response: Thank you, we agree that the field of literature is broad. In some cases we used reviews on common topics, but agree that citing additional original studies throughout strengthens the review article.

Reviewer 2 Report

This is a well written and very interesting review on cellular senescence in post-mitotic brain cells. This topic is new and is rarely described/understood in the field of neuroscience.

I have only few comments:

1) Figure 1b, the part illustrating the cell cycle is too small and very difficult to read

2) A Table summarizing the different senescence markers measured in the different brain cell types would be useful to the reader

3) To my point of view, the only missing key point of the review is the "functional" aspect of the presence of senescent cells in the brain: what are the consequences of neuronal/astrocytes/endothelial cell senescence on synaptic function? on cognition?

In this context, the authors should quote the study of Bussian et al (Nature 2018; 562:578–582), the first to show that elimination of senescent glial cells prevented cognitive decline in mice.

4) line 571, "the estimated upper limit net turnover rate is 0.001% per hour [116]"; this should be interpretated with caution. The in vivo turnover of endothelial cell is unknown and should not be extrapolated from a tissue-engineered model.

Author Response

1. “Figure 1b, the part illustrating the cell cycle is too small and very difficult to read”

Author response: We have modified the Figure 1b to make the cell cycle illustration bigger. We have also slightly increased all font sizes for Figure 1b for better readability. Please let us know if additional changes are needed – happy to make edits to ensure readability of our figure.

2. “A Table summarizing the different senescence markers measured in the different brain cell types would be useful to the reader”

Author response: We agree that a table would be useful to the reader. We have added a table to highlight key studies that were referenced or discussed in each section, organized by different biomarkers used to verify senescence.

3. “To my point of view, the only missing key point of the review is the "functional" aspect of the presence of senescent cells in the brain: what are the consequences of neuronal/astrocytes/endothelial cell senescence on synaptic function? on cognition? In this context, the authors should quote the study of Bussian et al (Nature 2018; 562:578–582), the first to show that elimination of senescent glial cells prevented cognitive decline in mice.”

Author response: This is a great suggestion by the reviewer. We have highlighted the important functional aspect of the presence of senescent cells in line 501 and 757 in addition to existing lines about observed association with cognition, synaptic function, demyelination, and integrity of BBB in lines 395, 404, 574, 630, 672, and 706.

4. “Line 571, "the estimated upper limit net turnover rate is 0.001% per hour [116]"; this should be interpretated with caution. The in vivo turnover of endothelial cell is unknown and should not be extrapolated from a tissue-engineered model.”

Author response: We appreciate this feedback and agree that there is a lack of physiological data for exact in vivo turnover rate of endothelial cells. We have removed this part of the sentence.

Reviewer 3 Report

Sah et al present a highly interesting review on an extremely important topic- the role of brain cell senescence in progressing neurodegenerative disorders. I found the review to be very informative and well organized.

I have a few comments and suggestions that I feel would strengthen the review:

  1. In lines 118, 173, 214 and 484, the authors should add published references from previous years, to the one mentioned, which has only been recently reviewed, and not published yet.
  2. Line 537- "Senescent astrocytes also release SASP factors that may induce surrounding cells to become senescent ". Please add reference to this statement.
  3. In lines 546-569, the introduction paragraph reviewing endothelial cell-senescence in general, as well as the phenomena in the periphery, is too long. Brain endothelial cells should be the primary concern.
  4. Lines 575-577: "The BBB is a highly selective semipermeable barrier with tight junctions that closely regulates the biochemical composition of the brain by restricting the free diffusion of nutrients, hormones, and pharmaceuticals". I suggest emphasizing that restriction of free diffusion is done by sealing of the paracellular space with tight junctions, which forces molecular traffic to take place through the endothelial membrane, where special routs of transport regulate it.
  5. Lines 580-584: "Recent studies suggest that brain endothelial cell senescence could contribute to BBB dysfunction though neurovascular uncoupling  and reactive oxygen species. Indeed, increased BBB permeability and vascular dysregulation have been observed in patients with early cognitive dysfunction, cerebral  microvascular diseases, and AD".  I suggest discussing whether BBB dysfunction, is part of a cascade of events which can induce endothelial cell senescence. BBB dysfunction can occur from early onset of brain insult, and may be induced by neuronal hyperexcitability, specifically seizures, which have been observed in patients with neurodegenerative disorders (AD, stroke, etc.). Molecular signalling which occurs from early onset of brain insult, can induce early BBB dysfunction, which is then followed by delayed BBB dysfunction, featuring overexpression of MMP’s, which have been suggested to induce cell senescence.
  6. Lines 605-606: "SASP factors may promote neuroinflammation and potentially affect the integrity of the BBB". Please add reference to this statement.
  7. Lines 648-651: "For instance, loss of OLs can lead to demyelination and is seen in Multiple Sclerosis (MS). Additionally, DNA damage is a known mediator of senescence suggesting a potential relationship between senescence of oligodendrocytes and neurodegenerative disorders". How does the fact that DNA damage mediates cell senescence explain the linkage between senescence of oligodendrocytes and neurodegeneration? Please explain. 

Author Response

1. “In lines 118, 173, 214 and 484, the authors should add published references from previous years, to the one mentioned, which has only been recently reviewed, and not published yet.”

 Author response: We have made references to Gillispie et al as our manuscript serves as a sister review on brain cell senescence. In order to avoid redundancy of references and details, we have referred to our review on mitotic cells for overlapping information. Gillispie et al has been accepted for publication and is available online now: https://www.mdpi.com/2075-1729/11/2/153. We have modified “in revision” to “in press,” and added additional references, as suggested by the referee, to accompany our review.

2. “Line 537- "Senescent astrocytes also release SASP factors that may induce surrounding cells to become senescent". Please add reference to this statement.”

Author response: We appreciate the referee’s suggestion. We have added references to line 590, which was reworded.

3. “In lines 546-569, the introduction paragraph reviewing endothelial cell-senescence in general, as well as the phenomena in the periphery, is too long. Brain endothelial cells should be the primary concern.”

Author response: We have removed a few details from the introduction paragraph that is not specific to the brain endothelial cells.

4. “Lines 575-577: "The BBB is a highly selective semipermeable barrier with tight junctions that closely regulates the biochemical composition of the brain by restricting the free diffusion of nutrients, hormones, and pharmaceuticals". I suggest emphasizing that restriction of free diffusion is done by sealing of the paracellular space with tight junctions, which forces molecular traffic to take place through the endothelial membrane, where special routs of transport regulate it.”

Author response: We appreciate this feedback and agree this is a great point we wish to emphasize in section 5. We have emphasized this statement in line 625.

5. “Lines 580-584: "Recent studies suggest that brain endothelial cell senescence could contribute to BBB dysfunction though neurovascular uncoupling and reactive oxygen species. Indeed, increased BBB permeability and vascular dysregulation have been observed in patients with early cognitive dysfunction, cerebral microvascular diseases, and AD".  I suggest discussing whether BBB dysfunction, is part of a cascade of events which can induce endothelial cell senescence. BBB dysfunction can occur from early onset of brain insult, and may be induced by neuronal hyperexcitability, specifically seizures, which have been observed in patients with neurodegenerative disorders (AD, stroke, etc.). Molecular signaling which occurs from early onset of brain insult, can induce early BBB dysfunction, which is then followed by delayed BBB dysfunction, featuring overexpression of MMP’s, which have been suggested to induce cell senescence.”

Author response: This is another great suggestion by the reviewer and adds to our statement in the introduction paragraph for section 5: “The co-occurrence of senescent endothelial cells with aging and disease makes it difficult to discern whether they are upstream mediators or downstream consequences of diseases.” We have addressed this relationship from line 634 to 640.

6. “Lines 605-606: "SASP factors may promote neuroinflammation and potentially affect the integrity of the BBB". Please add reference to this statement.”

Author response: We appreciate the referee’s suggestion. We have added references to support this statement about how SASP factors may promote neuroinflammation and affect the integrity of the BBB.

7. “Lines 648-651: "For instance, loss of OLs can lead to demyelination and is seen in Multiple Sclerosis (MS). Additionally, DNA damage is a known mediator of senescence suggesting a potential relationship between senescence of oligodendrocytes and neurodegenerative disorders". How does the fact that DNA damage mediates cell senescence explain the linkage between senescence of oligodendrocytes and neurodegeneration? Please explain.”

Author response: This suggestion by the reviewer is valid and appreciated. We have reordered the statements and added another sentence in this section to better reflect how cell senescence mediated by DNA damage explains the linkage between senescent OLs and neurodegeneration. We are suggesting that OLs become senescent as a result of DNA damage, lead to defective myelination, and could in turn bring about neurodegeneration. We hope that these modifications clarify this.

Round 2

Reviewer 3 Report

All of my comments were addressed in full.